# The Effects of Viral Structural Proteins on Acidic Phospholipids in Host Membranes

**DOI:** 10.3390/v16111714

**Published:** 2024-10-31

**Authors:** Ricardo de Souza Cardoso, Akira Ono

**Affiliations:** Department of Microbiology and Immunology, The University of Michigan, Ann Arbor, MI 48109, USA; rdsc@umich.edu

**Keywords:** influenza A virus (IAV), human immunodeficiency virus (HIV-1), acidic phospholipids, phosphatidylinositol (4.5)-bisphosphate (PI(4,5)P2), phosphatidylserine (PS), virus assembly, plasma membrane, lipid microdomains, IAV matrix protein-1 M1, IAV hemagglutinin, HIV-1 Gag

## Abstract

Enveloped viruses rely on host membranes for trafficking and assembly. A substantial body of literature published over the years supports the involvement of cellular membrane lipids in the enveloped virus assembly processes. In particular, the knowledge regarding the relationship between viral structural proteins and acidic phospholipids has been steadily increasing in recent years. In this review, we will briefly review the cellular functions of plasma membrane-associated acidic phospholipids and the mechanisms that regulate their local distribution within this membrane. We will then explore the interplay between viruses and the plasma membrane acidic phospholipids in the context of the assembly process for two enveloped viruses, the influenza A virus (IAV) and the human immunodeficiency virus type 1 (HIV-1). Among the proteins encoded by these viruses, three viral structural proteins, IAV hemagglutinin (HA), IAV matrix protein-1 (M1), and HIV-1 Gag protein, are known to interact with acidic phospholipids, phosphatidylserine and/or phosphatidylinositol (4,5)-bisphosphate. These interactions regulate the localization of the viral proteins to and/or within the plasma membrane and likely facilitate the clustering of the proteins. On the other hand, these viral proteins, via their ability to multimerize, can also alter the distribution of the lipids and may induce acidic-lipid-enriched membrane domains. We will discuss the potential significance of these interactions in the virus assembly process and the property of the progeny virions. Finally, we will outline key outstanding questions that need to be answered for a better understanding of the relationships between enveloped virus assembly and acidic phospholipids.

## 1. Introduction

One feature distinguishing enveloped from non-enveloped viruses is their absolute reliance on host cell membranes for particle assembly and release. During infection, some enveloped viruses utilize the membranes of intracellular compartments for assembly and budding, examples being dengue virus [1,2,3], oropouche virus [4,5], herpes simplex virus-1 [6], hepatitis B virus [7], and SARS-CoV-2 [8]. Others, such as respiratory syncytial virus [9,10,11], influenza virus [12,13,14], ebola virus [15,16,17], and human immunodeficiency virus-1 [18,19,20], use the plasma membrane as a platform for assembly. Notably, the interaction of structural proteins from these viruses with the plasma membrane could induce unique changes in the membrane microenvironment. While many studies of the relationships between viral replication and lipids focus on cholesterol and lipid rafts, an increasing number of studies have revealed that a diverse range of viruses from various families utilize or modulate acidic phospholipids to enhance virus replication, assembly, and budding.

This review will focus on the interactions between cellular acidic phospholipids and the structural proteins of two enveloped viruses that assemble at the plasma membrane, influenza A virus (IAV) and human immunodeficiency virus type 1 (HIV-1). Specifically, we will discuss how IAV hemagglutinin (HA; for the purpose of this review, HA will be regarded as a structural protein), IAV matrix protein 1 (M1) (Figure 1a), and HIV-1 Gag (Figure 1b) interact with the plasma membrane and influence phospholipid dynamics. We will then discuss how the virus-induced reorganization of acidic phospholipid could modulate the virus assembly process or the properties of nascent progeny virions, ultimately favoring or preventing the infection cycle. Lastly, we will illuminate critical open questions regarding the association of HA, M1, and Gag with acidic phospholipids that can guide future investigations in the field of virus-membrane interactions.

## 2. Classification, Structure, and Cellular Distribution of Phospholipids

Phospholipids constitute a class of lipids that can be subdivided into glycerophospholipids and sphingomyelins (SM) [21,22]. Glycerophospholipids, which will be the main focus of the present review, have hydrophobic acyl chains at the sn-1 and -2 positions and a phosphate-containing head group at the sn-3 position of the glycerol backbone [23] (Figure 1c). Since their initial discovery in the early 1800s [24], various phospholipids have been identified, including the zwitterionic lipids, phosphatidylcholine (PC) and phosphatidylethanolamine (PE), as well as the negatively charged lipids, phosphatidic acid (PA), phosphatidylserine (PS) (Figure 1d), and phosphatidylinositol (PI). The phosphatidylinositol lipids with a phosphorylated inositol headgroup (polyphosphoinositides) play diverse roles in cellular functions, which vary depending on the position(s) and the number of the phosphate group on the inositol ring. Accordingly, they are distinguished as PI(3)P, PI(4)P, PI(5)P, PI(3,4)P2, PI(3,5)P2, PI(4,5)P2 (Figure 1e), and PI(3,4,5)P3 [25].

The intracellular distribution of the phospholipids varies between different organelles and within a single organellar membrane, thereby regulating the localization of lipid-binding proteins and hence, their specific functions [26,27,28]. The plasma membrane, a longitudinally non-homogeneous and transversally asymmetric organelle [27,29,30,31], is particularly enriched in negatively charged phospholipids [27] such as PA, PS, and PI(4,5)P2, which at physiological pH, have negative net charges of −1/−2, −1, and −4, respectively [25,26,27,32,33,34]. These acidic phospholipids interact with cations and/or proteins in a headgroup-specific and/or charge-dependent manner (Figure 1d,e), promoting membrane reorganization via protein/lipid clustering and membrane curvature [33,35,36,37,38]. Additionally, some of them, specifically polyphosphoinositides, are essential for numerous cellular signaling processes, such as the regulation of actin cytoskeleton dynamics and organization [25]. Importantly, acidic phospholipids play a pivotal role not only in cell biology but also in the assembly and egress of many viruses [39], wherein defects in lipid–viral protein interactions can culminate in the failure of efficient virus spread, as demonstrated for the Ebola [40,41,42,43,44,45], Nipah, and measles viruses [46], along with the viruses that we will discuss in this review, i.e., influenza A virus and HIV-1.

## 3. The Functions and Lateral Distributions of Plasma Membrane Acidic Phospholipids

An important feature of the plasma membrane is its high negative charge [47,48,49]. Experiments using polybasic probes demonstrated that the more positively charged the probe is, the more it binds to the plasma membrane relative to the intracellular membranous compartments [48]. The same study demonstrated that even after depletion of more highly charged PI(4,5)P2 or PI(3,4,5)P3, a subset of a polybasic probe binds to the plasma membrane, which contains PS, as detected by the lactadherin (LactC2) probe. The rest of the polybasic probes relocated to the intracellular compartments enriched in PS. These findings indicate that PS contributes to the negative charge of the plasma membrane, and that this charge contribution promotes the localization of positively charged proteins [48]. Because of its abundance at the plasma membrane and its negative charge (Figure 1d), PS has been suggested to play a role in the diverse cellular functions via its contribution to protein recruitment and clustering [26,48,50,51]. The cellular functions to which PS contributes include endo-, exo-, and phagocytosis and cell polarity maintenance [52,53,54,55,56,57].

Another acidic phospholipid that contributes to the negative charge of the plasma membrane is PI(4,5)P2 [58]. PI(4,5)P2 localization and maintenance at the plasma membrane is mediated by phosphoinositide kinases and phosphatases, inter-organellar trafficking via vesicles or membrane contact sites, and protein–lipid interactions [25,59,60,61,62,63,64].

The interaction between PI(4,5)P2 and proteins can be solely charge-dependent, as exemplified by myristoylated alanine-rich C-kinase substrate (MARKCS), which interacts with the acidic headgroup of the PI(4,5)P2 molecule (Figure 1e) via the positively charged effector domain [65,66,67,68,69,70]. Alternatively, the PI(4,5)P2 binding of a protein can be mediated by a combination of both structural- and charge-dependent interactions, as seen for some proteins containing the pleskstrin homology domain (PH) [69,71,72,73,74,75]. Other representative protein domains that associate with PI(4,5)P2 include Bin Amphiphysin Rvs (BAR) [69,76,77] and Epsin/AP180 N-Terminal Homology (ENTH/ANTH) domains [78,79], which are necessary for cellular functions, such as clathrin-mediated endocytosis [78,79].

Indeed, PI(4,5)P2 regulates clathrin-mediated endocytosis [60], from its initiation to its completion, by recruiting multiple proteins at multiple steps [73,80]. Subunits of the clathrin adapter protein complexes interact, via their basic residues, with PI(4,5)P2, which triggers their conformational changes and subsequent membrane deformation, leading to the coated pit formation [81]. Interestingly, this interaction can stimulate phosphatidylinositol phosphate kinases (PIPKs) to produce a PI(4,5)P2 pool dedicated to clathrin endocytosis [82]. Interactions of PI(4,5)P2 with the PH domain of dynamin, a GTPase, are essential to cause the enrichment of dynamin in the budding membrane necks, which promotes pinching off and the release of the endocytic vesicle [80].

Local PI(4,5)P2 enrichment can occur through additional mechanisms other than the PIPKs stimulation mentioned above. Studies have shown that the phagocytosis event is marked by the accumulation of PI(4,5)P2, which induces actin nucleation and rearrangement at the phagosomal cup, and by subsequent redistribution of PI(4,5)P2 to other plasma membrane areas [53,83,84,85,86]. A study using fluorescence correlation spectroscopy and fluorescence recovery after photobleaching approaches revealed that the retention of PI(4,5)P2 is more likely due to molecular fences that restrict the escape of locally synthesized PI(4,5)P2 than to reduced diffusion inside the phagosomal cup [85].

Fusion, the final step in the exocytosis mechanism, also involves PI(4,5)P2 as a major player [87]. The fusion process is mediated by the SNARE complex, some of which consist of Syntaxin-1 and SNAP-25 [87]. Syntaxin-1, a transmembrane protein that localizes at the plasma membrane of the cells, has a juxtamembrane polybasic sequence (JMPBS) essential for the interaction with PI(4,5)P2 [88,89,90]. Interestingly, this interaction is able to induce the co-clustering of syntaxin-1 and PI(4,5)P2, which is distinguishable from the rest of the membrane [91].

In the cellular processes described above, PI(4,5)P2 serves as a molecule that recruits effector proteins to the specific site of the plasma membrane and often does so through local enrichment. Therefore, whether the proteins are recruited to pre-existing PI(4,5)P2 clusters or whether PI(4,5)P2 clusters are a result of their recruitment by proteins has been intensely studied [33,37,92,93]. Notably, two-thirds of plasma membrane PI(4,5)P2 is likely reversibly bound to proteins in the cells. This is supported by the significantly lower diffusion coefficient for fluorescent PI(4,5)P2 in the inner leaflet of the plasma membrane versus that in the protein-free liposome membrane [94]. Corroborating this possibility, PI(4,5)P2 can be found as clusters due to its association with basic residues of proteins [68,91,95,96].

As observed for host cellular proteins, viral structural proteins can alter the distribution of acidic phospholipids during virus assembly, whereas the distribution of acidic phospholipids can determine the site of virus assembly and the nature of nascent virus particles. Below, we will address the interplay between the acidic phospholipids and virus assembly, with a focus on the assembly of influenza and HIV virus particles.

## 4. Influenza A Virus (IAV) Assembly and Acidic Phospholipids

### 4.1. Influenza A Virus Assembly

The assembly of the influenza A virus (IAV) takes place at the plasma membrane of the infected cells. This process is orchestrated by its structural proteins, hemagglutinin (HA), neuraminidase (NA), matrix protein 1 (M1) (Figure 1a), matrix protein 2 (M2), and viral ribonucleoprotein complexes (vRNPs) [12,97,98]. The type I transmembrane protein hemagglutinin (HA), which is initially synthesized as the precursor HA0 [99], and the neuraminidase (NA), a type II transmembrane protein, reach the IAV assembly sites at the apical plasma membrane through the secretory pathway [12]. At the assembly sites, the acylation of the HA cytoplasmic tail promotes both HA-M1 interaction and membrane curvature at the assembly/budding sites [100]. NA plays a crucial role as a sialidase, cleaving the virus receptor sialic acid to facilitate the release of nascent IAV particles [101].

The M1 protein facilitates the export of the viral ribonucleoprotein complexes (vRNPs) from the nucleus to the cytosol [98,102]. At the plasma membrane, M1 forms a lattice underneath the lipid bilayer, forming the viral envelope, where it mediates the association of the viral transmembrane proteins, HA, NA, and M2, with the vRNPs [103]. Notably, the trafficking of M1 to the assembly sites depends on the presence of M2; the absence of M2 results in M1 mislocalization [104].

The assembly of HA, NA, M1, M2, and vRNPs complexes into nascent virus particles is coupled to particle budding at the plasma membrane [97]. The precise molecular mechanisms underlying the influenza budding steps are yet to be fully elucidated. However, several non-mutually exclusive mechanisms may contribute to the formation of membrane curvature, which include: (1) influenza protein accumulation, which bends the membrane, either via molecular crowding or through forming a curve-inducing molecular shape; (2) the accumulation of cone-shaped lipids in the inner leaflet of the assembly sites; and (3) the involvement of the cytoskeleton or other host proteins that alter the membrane structures [97]. Unlike HIV or many other enveloped viruses, which utilize the endosomal sorting complex required for transport (ESCRT) machinery for the release of virus particles, the release of influenza virus particles from the plasma membrane occurs in an ESCRT-independent manner [105]. During the late events in assembly, M2 protein localizes to the neck of the budding particle, facilitating the budding/pinching off process to release the virus.

The IAV infection modulates the cellular lipid metabolism [106,107,108,109,110,111]. In addition, it has long been known that during the IAV assembly, the plasma membrane gives rise to specialized membrane microdomains known as budozones [12,112,113,114,115]. The budozones are thought to consist of lipid raft-like microdomains enriched in cholesterol and sphingolipids where several IAV proteins associate with it [116,117]. Early evidence has shown that HA [118], NA [119,120], and M1 [121] partition into detergent-resistant membranes (DRM), which is consistent with, but not the indication of, lipid raft association. In contrast, M2 is thought to localize at the boundary between the raft and non-raft domains [12,122]. Consistent with this notion, immuno-electron microscopy of the plasma membrane showed that M2 is present outside of the HA-rich domains [123]. The IAV infection increases the lipid packing of the infected cell plasma membrane and causes lower lateral diffusion of membrane-associated proteins [124]. Therefore, the IAV infection causes changes in the membrane dynamics at the whole cell and the membrane microdomain levels, both of which could modulate assembly and budding [108,114,117,124,125,126,127].

Accumulating evidence has shown that in addition to the lipid raft-like microdomains, acidic phospholipids are host determinants regulating IAV assembly and budding [128], but they are also modulated by infection [124]. In the next section, we will focus on how IAV HA and M1 proteins interact with and modulate the acidic phospholipids during IAV assembly.

### 4.2. The Interplay Between Acidic Phospholipids and Influenza A Virus Structural Proteins

#### 4.2.1. Hemagglutinin (HA) and the Acidic Phospholipids

Fluorescence photoactivation localization microscopy (FPALM) of both living and fixed cells has revealed the presence of dynamic HA clusters at the plasma membrane, spanning in size from a few nanometers to a micrometer scale [129]. As alluded to earlier, it has been proposed that the lateral distribution of HA at the plasma membrane is determined by HA partitioning into cholesterol-enriched raft-like microdomains [112,123]. In support of this possibility, immunoelectron microscopy and FRET experiments showed that cholesterol depletion disrupts HA clustering and HA–raft marker association [123,130,131]. However, despite the cholesterol enrichment in IAV virions [117,125], a high-resolution secondary ion mass spectrometry (SIMS) study did not find a similar distribution of cholesterol inside and outside of the HA clusters at the plasma membrane [132]. Therefore, the exact role of cholesterol and the cholesterol-enriched microdomains in the formation of HA clusters remains to be elucidated [133].

Besides cholesterol, acidic phospholipids, such as PS and PI(4,5)P2, can be found in the membrane fractions that were thought to represent lipid rafts [134,135,136], and hence, it is conceivable that these lipids may play a role in HA partitioning/clustering at the plasma membrane. Consistent with the possibility that HA associates with PI(4,5)P2 at the plasma membrane, super-resolution experiments have shown that the motility of PI(4,5)P2 decreases in cells expressing the influenza protein, and that elevated HA expression correlates with increased PI(4,5)P2 clustering [137,138].

It was further shown that cetylpyridinium chloride (CPC), containing a monocationic head group, decreases HA clustering at the plasma membrane and reduces IAV loads and mortality in zebrafish [128]. CPC appears to act on PI(4,5)P2, as FPALM experiments demonstrate a decreased association between HA and a PI(4,5)P2 probe (GFP-tagged PH domain) in the presence of CPC. Additionally, the PM displacement of MARCKS, a PI(4,5)P2-binding protein, was observed in the presence of CPC [128]. These data collectively indicate that CPC abrogates the interaction between PI(4,5)P2 and HA. Altogether, the accumulating evidence suggests that PI(4,5)P2 may play a pivotal role in HA clustering, setting the stage for virus assembly and budding.

At present, it remains to be determined whether HA interacts directly or indirectly with acidic phospholipids. In vitro evidence favoring a possible direct association includes circular dichroism studies that demonstrated an increase in the helicity of an HA transmembrane peptide in membrane bilayers containing negatively charged lipids compared to the results for bilayers composed solely of zwitterionic lipids like PC [139]. The juxtamembrane C-terminal region in the HA protein of some IAV strains contains lysine or arginine residues [139,140] (Figure 2a), and since HA protein forms trimers, this can increase the positive charge in its cytoplasmic tail, potentially facilitating interactions with acidic phospholipids such as PS and PI(4,5)P2 (Figure 2b). Consistent with this possibility, a recent all-atom molecular dynamics (AAMD) simulation showed interactions between arginines in the HA cytoplasmic tail and phosphate groups in the PI(4,5)P2 head group [141]. Alternatively or additionally, HA may interact with PI(4,5)P2 indirectly through other proteins or protein complexes such as the actin cytoskeleton, which is regulated by PI(4,5)P2 and implicated in HA plasma membrane clustering [142].

#### 4.2.2. Matrix Protein-1 (M1) and the Acidic Phospholipids

The N-terminal region of the M1 protein is a globular structure consisting of nine alpha-helices, and some of them contain basic residues that favor its electrostatic association with phospholipids [143,144,145,146,147,148,149,150] (Figure 1a). In addition, due to its multimerization property [12,151,152,153], M1 protein can increase its positively charged surface per oligomer, thereby increasing its affinity for plasma membrane acidic phospholipids. Several studies, which rely mostly on bulk biochemical approaches, have shown that M1 associates with PS [133,147,153]. Studies using FRET and small-angle X-ray scattering (SAXS) have demonstrated that the M1 protein can increase or stabilize the clustering of PS in small unilamellar vesicles (SUVs) [154,155]. Consistent with the in vitro studies, a fluorescent protein fusion of M1 and the fluorescent analogue of PS were observed to colocalize at the plasma membrane in a locally concentrated manner in IAV-infected cells [155]. PS was observed to cluster, even in non-infected cells. Therefore, in these experiments, the fluorescent M1 derivative might have been targeted to pre-existing PS clusters. At this time, it remains to be determined whether M1 multimerization restricts the lateral diffusion of PS at the plasma membrane, thereby stabilizing the PS-enriched domain. Additionally, the effects of M1 in cells were not examined in the absence of other viral components, leaving the possible roles for other IAV proteins (e.g., HA and M2) in the M1-PS coclustering open.

Even though the effects of M2 on membrane curvature have been intensely studied, studies conducted in vitro using giant unilamellar vesicles (GUVs) have shown that M1 can also cause membrane curvature in the presence of acidic phospholipids [151,156]. This process is reportedly driven by charge rather than by lipid headgroup specificity, since PS, PG, and PI(4,5)P2 were all described to support the M1-dependent membrane deformation [151,156].

X-ray crystallography and cryo-electron tomography studies determined that the N-terminal domain of M1 protein contains a highly positively charged surface [144,157] (Figure 1a). The three arginine residues spanning the positions 76–78 in M1 are essential for M1 membrane binding and virus assembly [144,149]. Replacing the basic residues with neutral amino acids causes mislocalization of the M1 proteins in cells [149] and a defect in binding to liposomes containing acidic phospholipids, whether PS or PI(4,5)P2 [146]. Of note, recent studies have shown that M1 also interacts with PI(4,5)P2 in cells. Single molecule localization microscopy experiments demonstrated that M1 co-clusters with PI(4,5)P2; however, this co-clustering can be undone by incubating the cells with positively charged compounds like CPC [137]. Notably, co-clustering between M1 and HA at the plasma membrane can be disrupted by CPC, indicating that the PI(4,5)P2 clustering is important for the association between M1 and HA at the plasma membrane [137] (Figure 2b).

In addition to HA and M1, the nucleoprotein (NP), an essential component of vRNPs, was shown to bind to PI(4,5)P2 using in vitro binding assays. Additionally, the depletion of cellular PI(4,5)P2 was observed to prevent the binding of NP to the plasma membrane in both cells transiently expressing NP alone and cells infected with IAV [158]. Altogether, the studies outlined above have demonstrated that acidic phospholipids regulate multiple steps in IAV particle formation, including the proper distribution of M1 and NP (or potentially, vRNP) to the plasma membrane, M1 multimerization, HA-M1 co-clustering, and the generation of membrane curvature, thereby facilitating the efficient replication cycles.

## 5. Human Immunodeficiency Virus Type 1 (HIV-1) Assembly and Acidic Phospholipids

### 5.1. HIV-1 Assembly

In general, HIV-1 assembly takes place at the plasma membrane of infected cells, where the polyprotein Gag orchestrates the virus assembly and budding [18,19,20]. Gag is translated as a precursor polyprotein Pr55Gag, consisting of the matrix domain (MA), the capsid domain (CA), spacer peptide 1 (SP1), the nucleocapsid domain (NC), spacer peptide 2 (SP2), and the p6 domain (Figure 1b) [18]. The MA domain contains two signals that allow for Gag localization to the plasma membrane. First, MA receives a co-translational modification with a 14-carbon fatty acid, i.e., myristylation, on its N-terminal glycine, which provides MA with the capacity to interact with membranes via hydrophobic interactions [159,160,161,162,163,164,165,166,167,168]. Second, MA contains a conserved highly basic region (HBR), comprised of the positively charged amino acids lysine and arginine, which mediates Gag binding to the negatively charged phospholipid [18,161,162,169,170], in particular, PI(4,5)P2, as discussed below (Figure 1b). At the plasma membrane, Gag undergoes multimerization mediated by CA–CA and NC–RNA interactions [18]. The specific interactions of NC with full-length viral RNA is primarily responsible for genome packaging, while MA plays an important role in the incorporation of the HIV-1 envelope protein (Env) [18]. The p6 domain of Gag interacts with the ESCRT machinery to mediate the pinching off event [18]. After the particle release, Gag undergoes a series of viral protease-mediated cleavages, leading to maturation of the infectious particle [18,171].

### 5.2. The Interplay Between Cellular Phospholipids and HIV-1 Structural Protein Gag

#### 5.2.1. Gag and PI(4,5)P2

PI(4,5)P2 plays a pivotal role in HIV-1 assembly [18,172]. The depletion of cellular PI(4,5)P2, achieved by the expression of polyphosphoinositide 5-phosphatase IV, significantly impairs Gag localization to the plasma membrane, as well as particle assembly [173,174,175,176,177,178]. Moreover, the overexpression of a constitutively active Arf6, which leads to PI(4,5)P2 accumulation to the intracellular vesicles, redirects HIV-1 assembly and budding to these structures [173]. In addition, the inhibition of the Rab27-dependent trafficking of a kinase producing PI(4,5)P2 precursor to the plasma membrane [179], as well as knockdown of PI(4,5)P2-producing kinases [180], inhibits Gag localization to the plasma membrane. Altogether, these studies demonstrate the importance of PI(4,5)P2 in determining the location of HIV-1 assembly [172]. As for the interface for PI(4,5)P2 in Gag, genetic studies have revealed that the MA HBR of Gag is crucial for its interaction with PI(4,5)P2. Substitutions of basic residues within the HBR with neutral amino acids result in the mislocalization of Gag to the intracellular compartments and/or failure to bind any membranes [162,169,170,181,182]. Liposome binding and NMR experiments showed that the same HBR basic amino acid residues are important for PI(4,5)P2 interactions [178,182,183,184,185].

Although other phospholipids also contain negative charges on their phosphorylated headgroups, as it is the case for PS, in vitro lipid binding studies showed that the MA or Gag preferentially binds to PI(4,5)P2 in a manner that is not simply charge-dependent [178,186,187,188,189]. NMR-based studies also demonstrate that MA has a higher affinity for PI(4,5)P2 than for other phosphoinositides [183,190]. One NMR study using myristylated MA and a water-soluble PI(4,5)P2 analogue detected sequestration of the short 2′ acyl chain into the MA hydrophobic cleft [190], but this may be due to the use of the lipid with non-native short acyl chains [183]. A recent cryo-electron tomography study found that the mature Gag lattice showed a density consistent with the acyl chain sequestration of PI(4,5)P2 [191], although the identity of this density remains to be determined. Nonetheless, PI(4,5)P2 acyl chains, or rather, their saturation status, have been observed to affect Gag binding to lipid membranes [192].

Intriguingly, Gag MA has been shown to bind not only to PI(4,5)P2 but also to nucleic acids [193,194,195,196,197,198], and this MA–nucleic acid interaction can compete with MA’s binding to acidic lipids such as PS [182,199,200]. More specifically, in vitro liposome binding studies, including those performed in the presence of mammalian cell lysates, and cell-based Gag-RNA crosslinking studies collectively suggest that Gag utilizes cellular tRNAs to prevent MA HBR from binding to membranes that contain PS, which is ubiquitously present not only in the plasma membrane but also in the intracellular compartments [181,182,201,202,203,204,205,206,207]. Consistent with this notion, the interface in MA for tRNA largely overlaps with that for acidic phospholipids [208,209,210]. The current working model suggests that upon encountering PI(4,5)P2 at the plasma membrane, Gag replaces the MA HBR-bound tRNA with this lipid, thereby ensuring the localization to the plasma membrane [172]. Additionally, it is postulated that this mechanism may mediate the temporal regulation of Gag membrane binding [205,210].

Coarse-grained and long-timescale AAMD studies have shown that Gag and PI(4,5)P2 localize to the vicinity of each other in the membrane [211,212]. Moreover, stimulated emission depletion and fluorescence correlation spectroscopy have revealed that Gag can trap PI(4,5)P2 and cholesterol, but not phosphatidylethanolamine and sphingomyelin, around itself [213]. Lipidomics studies have demonstrated that the HIV-1 particle is enriched with PI(4,5)P2 compared to the composision of the plasma membrane of the producer cell [174,214]. These studies indicate that HIV assembly causes the relocalization and accumulation of PI(4,5)P2 to the particle assembly site in the plasma membrane. Due to its ability to multimerize via the CA and NC domains, membrane-bound Gag may form a lattice with large patches of basic residues facing the cytoplasmic leaflet of the plasma membrane, potentially promoting the enrichment or clustering of anionic lipids. In support of Gag multimer-dependent PI(4,5)P2 sequestration, in vitro studies using model membranes have demonstrated that Gag can induce the accumulation of PI(4,5)P2 around itself. Importantly, the highest accumulation of PI(4,5)P2 was observed when a full-length Gag was used, highlighting the contribution of multimerization to PI(4,5)P2 accumulation [215,216]. Interestingly, Gag also showed some preference for binding to pre-formed clusters of PI(4,5)P2 induced by cations like Ca2+ in studies performed in liposomes [215]. These results highlight Gag’s versatility, as it can both cause PI(4,5)P2 clustering and utilize pre-formed PI(4,5)P2 clusters, at least in vitro.

While PI(4,5)P2 enrichment in assembling and released virus particles is recognized (Figure 2c), whether or not it performs any physiological function remains unknown. Interestingly, TIRF STORM super-resolution studies performed in HeLa and T cells demonstrated that Gag HBR and the JMPBS of the host transmembrane proteins, CD43, PSGL-1, and CD44, are important for their co-clustering at the assembly sites [217]. Importantly, these host proteins are incorporated into HIV-1 particles, and they can act as pro- or anti-viral factors, altering the fate of released virus particles [218,219,220,221,222,223]. In support of a role for PI(4,5)P2 in host protein incorporation, we observed that PI(4,5)P2 promotes Gag co-clustering with, and virus incorporation of, CD43, PSGL-1, and CD44 [224]. Together, these observations suggest that PI(4,5)P2 at the assembly sites plays a role not only in facilitating HIV-1 particle assembly but also regulating the sorting of host factors into the released particle.

#### 5.2.2. HIV-1 and PS

While PI(4,5)P2 promotes HIV-1 assembly by interacting with Gag, the role of PS seems to be less clear. Liposome binding studies showed that the MA domain in Gag can bind to membranes containing PS [161,225]. An NMR study suggested that MA interacts simultaneously with PI(4,5)P2 and PS [226], although the latter apparently binds to a region outside of the HBR. An AAMD study using the membrane models containing both PI(4,5)P2 and PS showed that both anionic lipids interact with MA around the HBR [227]. However, as alluded to earlier, in the presence of cell lysates, this binding of Gag to PS is inhibited by the presence of RNA [228]. Therefore, the precise role PS plays in Gag localization to the plasma membrane remains to be determined.

Many viral infections, including HIV-1 infection, promote the exposure of PS to the outer leaflet of the infected cells [229,230]. One mechanism for this PS exposure on the host cell membrane is an apoptotic response induced by HIV-1. As a consequence, during the assembly, the virus incorporates this exposed PS into its particles [231]. Once incorporated into the particle envelope, PS appears to play a dual role in HIV-1 dissemination [229]. PS exposure on the outer membrane acts as an “eat me” signal [232,233], a mechanism exploited by the vaccinia virus [234,235], which facilitates contact with and internalization by potential target cells, contributing to viral spread. Likewise, PS exposure on the envelope facilitates HIV-1 infection by increasing virus binding to PS receptors on the target cell. Therefore, in this context, PS serves as a co-factor aiding in viral spread [231]. However, PS can also promote cellular antiviral activities. HIV-1 particles exposing PS can be trapped by the TIM family proteins expressed in virus producer cells [236] and hence, fail to spread to target cells. PS is also implicated in the antiviral function of a host multi-transmembrane protein, SERINC5 [237,238]. SERINC5 is incorporated into virions and acts as a restriction factor, primarily by altering Env conformation and suppressing its activity [239]. Although the precise mechanism by which SERINC5 impairs viral infectivity is still to be determined, several studies have shown that SERINC5 modulates PS on the viral envelope [239]. A recent study demonstrated that SERINC5 has the ability to disrupt the membrane asymmetry of the HIV-1 envelope by exposing PS, PC, and PE on the outer leaflet, and that the degree of PS exposure correlates with the reduction in infectivity [240]. A subsequent study confirmed the phospholipid scramblase-like activity of SERINC5, but observed that the externalized PS levels do not strictly correlate with the negative impact of SERINC5 on viral infectivity [241]. Altogether, these findings suggest that PS in the HIV-1 particles, especially in the outer leaflet of the envelope membrane, may prevent or facilitate HIV-1 infectivity and spread; however, the precise role played by PS in HIV-1 assembly and spread remains to be determined.

## 6. Future Directions

The studies on IAV and HIV-1 outlined above collectively support the pivotal roles played by the plasma membrane acidic phospholipids in assembly and budding. The interaction between the structural proteins of these enveloped viruses and acidic phospholipids affects the steps of the virus particle assembly process, such as the binding of the proteins to the plasma membrane, oligomerization of the proteins, and generation of membrane curvature. During this process, the viral structural proteins can dramatically alter the local organization of the plasma membrane. It is also likely that multiple lipids temporally or stably participate in promoting protein clustering, creating a conducive environment for viral assembly and budding.

Although the recent studies have advanced our understanding of the relationships between acidic phospholipids and IAV or HIV-1, several questions, which are often common regarding both viruses, remain to be addressed. The significance of these questions is not limited to the two viruses, since the assembly processes of other enveloped viruses, including filoviruses [40,41,42,43,44,45] and paramyxoviruses [46,242,243], which comprise major emerging or reemerging viruses, are now known to involve acidic phospholipids. We highlight several of the outstanding questions below.

(1)Are viral structural proteins recruited to pre-existing acidic phospholipid-rich areas, or do they cause acidic phospholipids clustering or both?

Biophysical and cell-based studies have shown that host cellular proteins, such as MARCKS [66,68,244], syntaxin-1 [90], and other proteins with positively charged sequences (Figure 2a,d), induce the clustering of acidic phospholipids around them. Therefore, it is likely that acidic phospholipid clusters pre-exist at the plasma membrane before viral structural proteins are expressed. However, it is unknown if these clusters serve as the site of virus assembly (Figure 2d).

In the case of HIV-1, the reduction of the PI(4,5)P2 diffusion rate at the Gag clusters and PI(4,5)P2 enrichment in the virions support the possibility that viral structural proteins can generate acidic phospholipid clusters at the assembly sites (Figure 2c). However, it remains to be determined whether the acidic phospholipid clusters formed at the assembly sites are stable in the time scale of the assembly process (i.e., minutes to tens of minutes) and whether these clusters have any function (see below).

(2)Do acidic phospholipids play a role in the incorporation of viral transmembrane proteins and the packaging of viral genomes into nascent virus particles?

The studies described above revealed that acidic phospholipids, in particular PI(4,5)P2, associate with both HA and M1 and promote HA-M1 co-clustering [128,137], which may facilitate the formation of the virus assembly sites (Figure 2b). Therefore, it is likely that PI(4,5)P2 promotes the incorporation of HA into nascent IAV particles. However, the mechanistic details remain to be determined. As for HIV-1, it remains to be determined whether acidic phospholipids are involved in the incorporation of viral glycoprotein Env into assembling virus particles, even though a polybasic surface has been identified in the Env cytoplasmic tail [245].

HIV-1 Gag binds both plasma membrane and viral genomic RNA and is therefore solely responsible for genome packaging during the process of particle assembly. In nascent IAV particles, however, vRNP complexes are bound to the M1 lattice but not to the lipid bilayer [246,247]. Therefore, even though PI(4,5)P2 was shown to be important for the plasma membrane localization of NP and genome packaging into virus particles [158], it remains unknown whether or how NP or vRNP switch the binding partner from PI(4,5)P2 to M1.

(3)Do acidic phospholipids regulate the recruitment of host cellular proteins to the assembly sites, and if so, what roles do these host proteins play in the assembly process or virion infectivity?

Our study has shown that the incorporation of CD43, PSGL-1, and CD44, which contain juxtamembrane polybasic sequences, into HIV particles was reduced up to 20-fold upon PI(4,5)P2 depletion [224], indicating that PI(4,5)P2 promotes the association between these host cellular transmembrane proteins and HIV-1 Gag (Figure 2c,d). It is conceivable that the mechanism by which these host transmembrane proteins are incorporated into HIV-1 particles may be analogous to the association between IAV M1 and HA at the plasma membrane during assembly (Figure 2b). If the acidic phospholipids attract host transmembrane proteins solely via electrostatic interactions, it is crucial to understand whether and how the viruses regulate the recruitment of a variety of transmembrane proteins with basic cytoplasmic sequences to the assembly sites, since these host transmembrane proteins could have pro-viral (e.g., CD44) or anti-viral (e.g., CD43 and PSGL-1) effects on virus infectivity or spread.

In addition to recruiting the host transmembrane proteins, the clustering of acidic phospholipids, in particular PI(4,5)P2, at the virus assembly sites, may recruit other host proteins that may facilitate or suppress specific steps in the virus assembly process (Figure 2c). For example, some membrane-modulating proteins (e.g., IRSp53 [248]) and the actin cytoskeleton are regulated by PI(4,5)P2 and have been shown to affect the virus assembly process [13,249,250].

(4)What is the role, if any, of the incorporated acidic phospholipids for IAV and HIV in viral spread?

The accumulation of acidic phospholipids at the assembly sites likely leads to the presence, if not enrichment, of the lipids in the nascent virus particles. While the studies discussed in this review and elsewhere [229,251,252] support the roles for PS incorporated into enveloped virus particles, the roles played by virion-associated PI(4,5)P2 during the early stages of the virus replication cycle, i.e., attachment and entry, are poorly understood. Notably, cryo-EM studies revealed that the maturation of the HIV-1 virions apparently changes the patterns of interactions between MA and PI(4,5)P2 [191], which led the authors to propose their possible signaling function upon fusion with the target cell membrane.

(5)How do viral proteins regulate acidic phospholipid distribution locally and globally?

To address the questions above, it will be essential to determine the molecular interfaces in the viral structural proteins and the contribution of their oligomerization to the acidic phospholipid distribution. In this review, we focused on the viral structural proteins, HA, M1, and Gag; however, it will be also important to examine whether other viral proteins play any roles, e.g., by locally or globally modulating cellular enzymes that favor the accumulation of acidic phospholipids at the viral assembly sites.

The questions listed above highlight some, but not all, areas in which further research is needed. The advancement of research methodologies including, but not limited to, structural, microscopic, and omics approaches, will allow for our deeper understanding of the intricate interplay between enveloped viruses and acidic phospholipids during infection, potentially illuminating new therapeutic strategies for controlling enveloped virus assembly and spread.

## Figures and Tables

**Figure 1 viruses-16-01714-f001:**
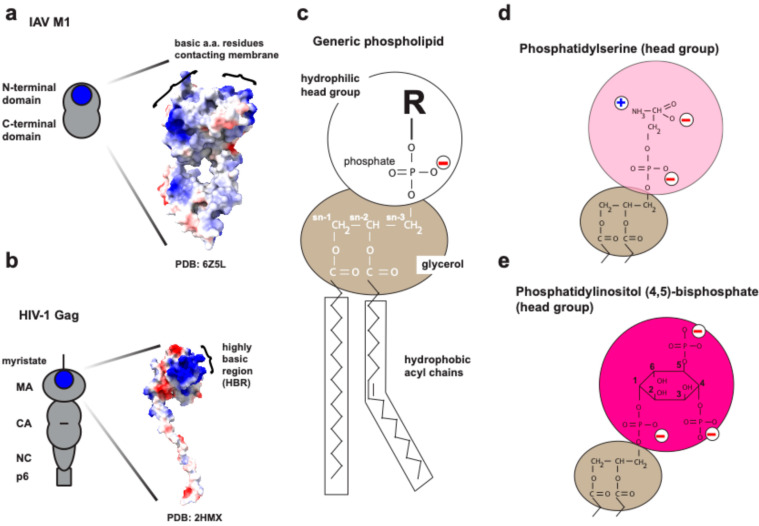
Structures of viral structural proteins and phospholipids. (**a**,**b**) Cartoon illustrations and structures of IAV M1 and HIV-1 Gag proteins. PDB accession numbers for the structures are shown at the bottom of each panel. Positively and negatively charged surfaces are indicated in blue and red, respectively, using UCSF ChimeraX software (v1.8). Basic residues involved in membrane binding are shown as blue circles in the cartoons and bracketed areas in the structures. (**c**) Schematic representation of a generic phospholipid structure. (**d**,**e**) Structures of the hydrophilic head group of phosphatidylserine (PS) and phosphatidylinositol (4,5)-bisphosphate [PI(4,5)P2)]. Circles with a “−” or “+” symbol indicate the charges present in the PS and PI(4,5)P2 head group.

**Figure 2 viruses-16-01714-f002:**
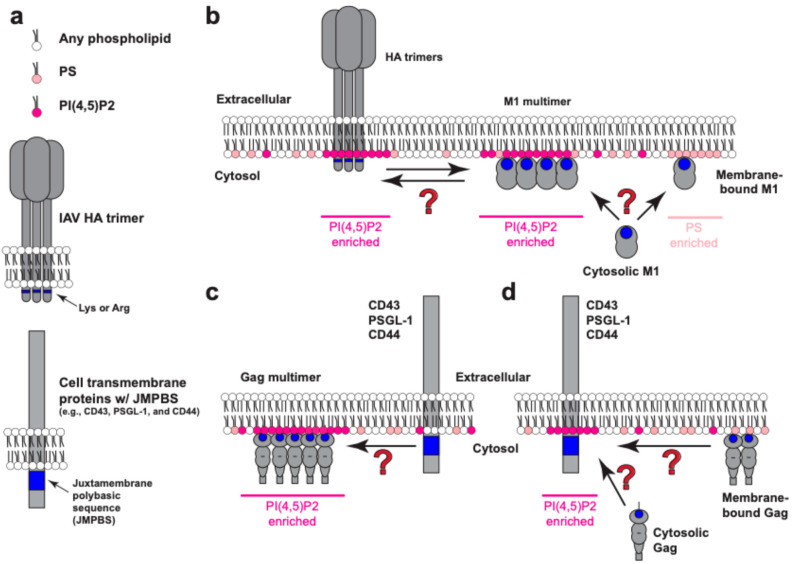
Potential mechanisms for viral and host transmembrane protein sorting at assembly sites mediated by acidic phospholipids. (**a**) Cartoon representation of phospholipids, IAV HA trimers, and host transmembrane proteins illustrating their association with the membrane. Basic residues of HA and the juxtamembrane polybasic sequences (JMPBS) are shown in blue. (**b**) HA trimers or M1 multimers associate with PI(4,5)P2-enriched membrane domains leading to the recruitment of each other. M1 also interacts with PS-enriched membrane domains, and this association may precede localization to the PI(4,5)P2-enriched domain. (**c**) HIV-1 Gag is localized in a PI(4,5)P2-enriched membrane domain, to which cellular transmembrane proteins with a JMPBS are recruited. (**d**) JMPBS of cellular proteins are present in a PI(4,5)P2-enriched membrane domain, which recruits cytosolic or plasma membrane-associated Gag.

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
