# Peer review of "The Effects of Viral Structural Proteins on Acidic Phospholipids in Host Membranes"

_viruses, 2024, doi:10.3390/v16111714_

Round 1
Reviewer 1 Report
Comments and Suggestions for Authors
This is an interesting and informative review on the role of acidic lipids in the assembly of IAV and HIV-1.
Regarding IAV, it might be interesting to mention whether NA also might interact with acidic lipids.
Also, what about the role of positively charged lipids in electrostatic interactions with proteins? Although I know that such lipids are very few...
Readers might appreciate information regarding the charge of the discussed lipids. Is that easy? At pH 7 ca., PS has a -1 charge, like e.g. PI3P, and PI45P has a -2 charge?
Regarding the paragraph starting on line 398-9. I guess many viruses, including IAV, induce apoptosis and PS exposure. So what is described here for HIV should also hold true for IAV, shouldn´t it?
Typos on line 214 and 251.
Line 253, there is no Figure 1F
Author Response
We thank the reviewers for the thorough assessment of our manuscript and helpful suggestions. Please see the attachment for our response to each of the reviewers' comments.

Reviewer 2 Report
Comments and Suggestions for Authors
NA

NA
Author Response

(The authors gave the same response as above.)
